# Effect of the alveolar recruitment maneuver during laparoscopic colorectal surgery on postoperative pulmonary complications: A randomized controlled trial

Yu Kyung Bae[1], Sun Woo Nam[2,3], Ah-Young Oh[1,3]*, Bo Young Kim[1], Bon-Wook Koo[1], Jiwon Han[2,4], Subin Yim[1,3]

1 Department of Anesthesiology and Pain Medicine, Seoul National University Bundang Hospital, Seongnam, Republic of Korea, 2 Department of Anesthesiology and Pain Medicine, Chung-Ang University Gwangmyeong Hospital, Gwangmyeong si, Republic of Korea, 3 Department of Anesthesiology and Pain Medicine, Seoul National University, College of Medicine, Seoul, Republic of Korea, 4 Department of Anesthesiology and Pain Medicine, Chung-Ang University, College of Medicine, Seoul, Republic of Korea

☯ These authors contributed equally to this work.
* ohahyoung@hanmail.net

**Data Availability Statement:** All relevant data are within the manuscript and its Supporting Information files.

## Abstract

Intraoperative lung-protective ventilation, including low tidal volume and positive end-expiratory pressure, reduces postoperative pulmonary complications. However, the effect and specific alveolar recruitment maneuver method are controversial. We investigated whether the intraoperative intermittent recruitment maneuver further reduced postoperative pulmonary complications while using a lung-protective ventilation strategy. Adult patients undergoing elective laparoscopic colorectal surgery were randomly allocated to the recruitment or control groups. Intraoperative ventilation was adjusted to maintain a tidal volume of 6–8 mL kg$^{-1}$ and positive end-expiratory pressure of 5 cmH$_2$O in both groups. The alveolar recruitment maneuver was applied at three time points (at the start and end of the pneumoperitoneum, and immediately before extubation) by maintaining a continuous pressure of 30 cmH$_2$O for 30 s in the recruitment group. Clinical and radiological evidence of postoperative pulmonary complications was investigated within 7 days postoperatively. A total of 125 patients were included in the analysis. The overall incidence of postoperative pulmonary complications was not significantly different between the recruitment and control groups (28.1% vs. 31.1%, $P = 0.711$), while the mean ± standard deviation intraoperative peak inspiratory pressure was significantly lower in the recruitment group (10.7 ± 3.2 vs. 13.5 ± 3.0 cmH$_2$O at the time of CO$_2$ gas-out, $P < 0.001$; 9.8 ± 2.3 vs. 12.5 ± 3.0 cmH$_2$O at the time of recovery, $P < 0.001$). The alveolar recruitment maneuver with a pressure of 30 cmH$_2$O for 30 s did not further reduce postoperative pulmonary complications when a low tidal volume and 5 cmH$_2$O positive end-expiratory pressure were applied to patients undergoing laparoscopic colorectal surgery and was not associated with any significant adverse events. However, the alveolar recruitment maneuver significantly reduced intraoperative peak inspiratory pressure. Further study is needed to validate the beneficial effect of the alveolar

**Funding:** The author(s) received no specific funding for this work.

**Competing interests:** The authors have declared that no competing interests exist.

recruitment maneuver in patients at increased risk of postoperative pulmonary complications.

**Trial registration:** Clinicaltrials.gov (NCT03681236).

## Introduction

Abdominal surgery is a non-modifiable risk factor for postoperative pulmonary complications (PPCs) [1, 2]. The incidence of PPCs after major surgery is as high as 48% [3]. In addition to atelectasis, which develops in 90% of patients during general anesthesia, a prolonged $CO_2$ pneumoperitoneum and the Trendelenburg position during surgery induce further cephalad displacement of the diaphragm, along with increased intrathoracic pressure, decreased lung compliance, decreased functional residual capacity, and impaired arterial oxygenation [4–6].

Lung protective ventilation was initially studied in patients with acute respiratory distress syndrome (ARDS). A low tidal volume, positive end-expiratory pressure (PEEP), and the alveolar recruitment maneuver (RM) improve survival [7]. The beneficial effect of lung-protective ventilation for surgical patients without severe underlying lung disease is still being investigated. Low tidal volume and PEEP seem to decrease PPCs, although the optimal PEEP is controversial [6, 8–11]. In contrast, the efficacy and the specific RM method for reducing PPCs have not been established.

The present study aimed to evaluate whether the intermittent intraoperative RM at three time points (the start and end of the pneumoperitoneum, and immediately before extubation) with a continuous positive airway pressure (CPAP) of 30 $cmH_2O$ for 30 s could further reduce PPCs in patients undergoing laparoscopic colorectal surgery using a lung-protective ventilation strategy, including low tidal volume 6–8 mL $kg^{-1}$ and a moderate PEEP 5 $cmH_2O$.

## Materials and methods

### Study design

This prospective randomized controlled study was approved by the Institutional Review Board of Seoul National University's Bundang Hospital (B-1708-415-302), Gyeonggi-do, Republic of Korea, and was registered at ClinicalTrials.gov. (NCT03681236, September 20, 2018). Written informed consent was obtained from all subjects.

Patients aged 18–70 years, scheduled for elective laparoscopic colorectal cancer surgery at the Seoul National University Bundang Hospital from February 2018 to February 2021, were included. The exclusion criteria were American Society of Anesthesiologists (ASA) physical status $\geq$ 3, history of severe cardiopulmonary disease, history of mechanical ventilation therapy within the last 6 months, and inability to provide informed consent.

### Randomization and blinding

Patients were randomly allocated to the recruitment or control groups at a 1:1 ratio using a set of computer-generated randomization codes (Random Allocation Software, ver. 1.0; Informer Technologies, Los Angeles, CA, USA) in sealed envelopes. Patients, postoperative care unit (PACU) nurses, ward nurses, and the investigator assessing the PPCs were blinded to the group assignments.

## Anesthesia protocol

Non-invasive blood pressure, three-electrode electrocardiogram, and pulse oximetry were applied as routine monitoring of the patients in the operating theatre.

Propofol 1.0–2.0 mg kg$^{-1}$ and target-controlled infusion of remifentanil 3.0 ng mL$^{-1}$ and rocuronium 0.5–1.0 mg kg$^{-1}$ were used to induce anesthesia. After intubation, anesthesia was maintained with desflurane 6–8 vol%, target-controlled infusion of remifentanil 0.1–3.0 ng mL$^{-1}$, and boluses of rocuronium 5–10 mg, as needed. Neuromuscular blockade was reversed in both groups with sugammadex 200–400 mg.

## Ventilation protocol

The ventilator was set to pressure-controlled mode after tracheal intubation. The fraction of inspired $O_2$ and tidal volume (TV) were 0.4 and 6–8 mL kg$^{-1}$, respectively, based on ideal body weight, and 5 cmH$_2$O PEEP was maintained until the end of surgery. The respiratory rate was adjusted so that the end-tidal partial pressure of $CO_2$ was 35–38 mmHg, with an inspiration-to-expiration ratio of 1:2.

The intraoperative peak inspiratory pressure (PIP) needed to maintain a TV of 6–8 mL kg$^{-1}$ was documented at the time of induction, $CO_2$ gas-in, $CO_2$ gas-out, and recovery. If the PIP was > 30 cmH$_2$O for > 5 min, the endotracheal tube and ventilator circuit were examined for mechanical faults (e.g., kinking of the endotracheal tube, or water or secretions in the circuit). The study was discontinued if the PIP remained > 30 cmH$_2$O after any mechanical issues had been addressed.

The RM was manually applied by maintaining a continuous positive pressure of 30 cmH$_2$O for 30 s three times in the recruitment group (at the start and end of the pneumoperitoneum and immediately before extubation).

## Outcome variables

The primary outcome was the incidence of PPCs within 7 days after surgery, which was defined as having at least one of the following respiratory symptoms or signs: hypoxia, suspected pulmonary infection, pleural effusion, atelectasis, or pulmonary infiltration. The $O_2$ saturation of the patient was monitored in the PACU. We defined hypoxia as SpO$_2$ < 90% or PaO$_2$ < 60 mmHg on room air. Severe hypoxia was defined as persistent hypoxia despite $O_2$ supplementation; the need for $O_2$ therapy when leaving the PACU was also documented. Postoperative body temperature was reviewed. If there was a fever (tympanic temperature > 37.5°C) without any other cause on day 2 after surgery, which resolved with lung care, it was assumed to be most likely due to atelectasis and was documented. Chest X-rays were routinely taken on day 2 after surgery and checked for abnormal findings, such as atelectasis, pleural effusion, and pulmonary infiltration. The findings of any additional chest images taken within 7 days after the surgery were also considered. Hypoxia, severe hypoxia, and suspected pulmonary infection were recorded within 7 days after surgery. Suspected pulmonary infection was defined as the use of antibiotics without another source of infection and fulfillment of at least one of the following criteria: new or changed sputum, new or changed lung opacities on chest X-ray, tympanic temperature > 38.3°C, or white blood cell count > $12 \times 10^9$ L$^{-1}$ [9].

The secondary outcomes were PIP and the calculated compliance: TV (mL) / (PIP–PEEP: Driving Pressure, DP) (cmH$_2$O), which were measured after inducing anesthesia, at the start and end of pneumoperitoneum, and upon recovery. The incidence of hypotension during the RM and the administration of a vasopressor was also assessed.

## Statistical analysis

The incidence of PPCs in patients undergoing laparoscopic surgery was estimated to be 30% based on a previous study [4]. A sample size of 69 patients in each group was calculated to detect a 20% difference in the incidence of atelectasis, with 5% type 1 error, 80% power, and 10% dropout rate based on chi-square test for comparing proportions.

Continuous variables are presented as the mean with standard deviation (SD) and the median with interquartile range (IQR), and categorical variables are presented as numbers with percentages. The groups were compared with Student's *t*-test or the Mann–Whitney *U*-test for continuous variables, and the chi-square test or Fisher's exact test for categorical variables. Analysis of variance (ANOVA) was conducted for the repeated variables. A linear mixed model (LMM) was used for the analysis if normality was not met for ANOVA. The compound symmetry covariance structure was used in the LMM and the criteria was default for selecting it. All statistical analyses were performed using SPSS ver. 24 software (IBM Corp., Armonk, NY, USA). A *P*-value $< 0.05$ was considered significant.

## Results

A total of 138 patients were eligible for the study. The procedure was converted to open surgery in one case, no postoperative chest X-ray data were available in one case, and non-compliance with the protocol was observed in 11 cases. After excluding these cases, 125 patients were analyzed (Fig 1).

The demographic and surgical data were not significantly different between the recruitment and control groups (Table 1).

Five patients (7.8%) in the recruitment group and nine (14.8%) in the control group were discharged from the PACU on supplemental $O_2$ ($P = 0.219$). The incidence of fever within 2 days postoperatively was not different between the recruitment and control groups (87.5% vs. 86.9%, $P = 0.918$). The incidence of PPCs was not different between the two groups in the PACU or on the ward. In one patient, more than one PPC was present. The number of PPCs observed in the PACU and the ward was measured by separating each symptom or sign that appeared in the same patient. However, when counting total PPCs, it was counted as one PPC if a patient had one or more PPCs (Table 2).

The intraoperative PIP needed to maintain a TV of 6–8 mL $kg^{-1}$ did not differ between the two groups at the time of anesthesia induction or $CO_2$ gas-in, but was significantly lower at $CO_2$ gas-out and recovery in the recruitment group. The calculated compliances were also significantly higher at the last two time points in the recruitment group. Intraoperative peak inspiratory pressure was significantly lower in the recruitment group ($10.7 \pm 3.2$ vs. $13.5 \pm 3.0$ cmH$_2$O at the time of $CO_2$ gas-out, $P < 0.001$; $9.8 \pm 2.3$ vs. $12.5 \pm 3.0$ cmH$_2$O at the time of recovery, $P < 0.001$). Intraoperative compliance was significantly higher in the recruitment group ($96.3 \pm 7.7$ vs. $59.0 \pm 3.0$ ml/cmH$_2$O at the time of $CO_2$ gas-out, $P < 0.001$; $121.5 \pm 10.3$ vs. $69.6 \pm 4.1$ ml/cmH$_2$O at the time of recovery, $P < 0.001$). (Fig 2).

The incidence of intraoperative hypotension during or immediately after the RM and the administration of a vasopressor was comparable between the groups (induction = 9 vs. 15, gas-in = 0 vs. 2, gas-out = 3 vs. 3, recovery = 2 vs. 4, $P > 0.05$). No significant difference was observed in the amount of vasoconstrictor used between the two groups (mean ephedrine dose, mg = 6.25 vs. 7.83, $P = 0.164$, mean phenylephrine dose, μg = 16.56 vs. 11. 95, $P = 0.184$).

The median [25th–75th IQR] length of the postoperative hospital stay was comparable between the recruitment and control groups (7.0 [5.0–8.0] vs. 7.0 [6.0–8.0] days, $P = 0.870$). Patients with a PPC had a longer median [25th–75th IQR] postoperative hospital stay than those who did not (7.0 [6.0–10.0] vs. 6.0 [5.0–8.0] days, $P = 0.002$).

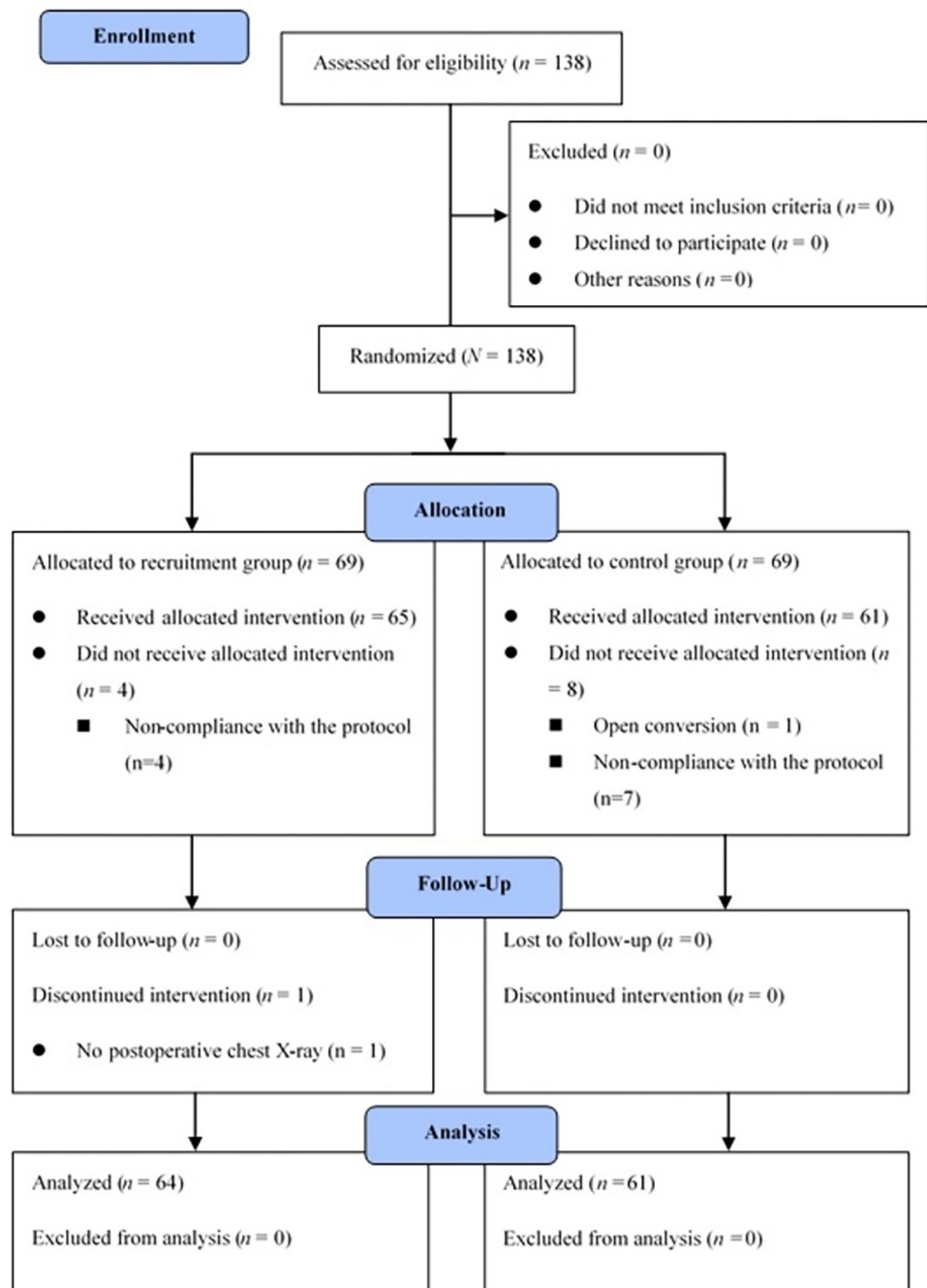

**Fig 1. CONSORT flow diagram.**

## Discussion

The overall incidence of PPCs was about 30% when using lung-protective ventilation, low TV 6–8 mL kg$^{-1}$, and moderate PEEP 5 cmH$_2$O in patients with no severe underlying lung disease. However, adding the RM did not lead to a further decrease in the incidence of PPCs.

**Table 1. Patient and surgery characteristics.**

|  | Recruitment group (*n* = 64) | Control group (*n* = 61) | *P*-value |
|---|---|---|---|
| Age, years | 63.0 (12.0) | 61.4 (12.9) | 0.472 |
| Sex, male/female | 39 (60.9%)/25 (39.1%) | 45 (73.8%)/16 (26.2%) | 0.127 |
| Height, cm | 163.4 (8.3) | 165.1 (8.7) | 0.270 |
| Weight, kg | 63.7 [53.8–72.7] | 66.6 [57.3–75.8] | 0.111 |
| BMI, kg m$^{-2}$ | 23.6 (3.2) | 24.3 (3.0) | 0.231 |
| ASA physical status, I/II | 17/47 | 11/50 | 0.253 |
| Duration of surgery, min | 135.0 [110.0–173.75] | 150.0 [120.0–190.0] | 0.184 |
| Duration of anesthesia, min | 182.5 [156.3–222.5] | 190.0 [160.0–240.0] | 0.317 |
| Duration of pneumoperitoneum, min | 99.0 [75.0–129.5] | 95.0 [72.5–140.0] | 0.943 |

Continuous values are shown as the mean (standard deviation) or median [25th–75th interquartile range]. Significance was assessed as *P* < 0.05. Categorical variables are expressed as the number of patients. ASA, American Society of Anesthesiologists.

PPCs increase the length of the hospital stay, intensive care unit (ICU) stay, and in-hospital mortality [12, 13]. In our study, the majority of the observed PPCs were atelectasis or minimal pleural effusion on chest X-ray, and no ICU admissions or deaths occurred during the postoperative hospital stay. Nevertheless, the length of the postoperative hospital stay was significantly longer in patients who developed a PPC than those who did not. The importance of preventing PPCs must be emphasized, given that even minor PPCs in relatively healthy patients with ASA physical status 1–2 can lead to a longer hospital stay and an increase in medical expenses.

In this study, the target TV was achieved intraoperatively in the recruitment group by a significantly reduced PIP and DP, suggesting that the RM alleviated the atelectasis caused by general anesthesia and the pneumoperitoneum. This result is consistent with previous reports that the RM improves intraoperative lung mechanics by opening the alveoli and increasing lung compliance [14]. The current study was aimed at patients without underlying lung disease, and reducing PIP did not affect the incidence of PPCs. However, in a large-scale multicenter

**Table 2. Incidence of postoperative pulmonary complications.**

|  | Recruitment group (*n* = 64) | Control group (*n* = 61) | *P*-value |
|---|---|---|---|
| *PACU* |  |  |  |
| Hypoxia | 4 (6.3) | 3 (4.9) | 1.000 |
| Severe hypoxia | 0 (0) | 0 (0) | - |
| *Ward* |  |  |  |
| Hypoxia | 4 (6.3) | 4 (6.6) | 1.000 |
| Severe hypoxia | 0 (0) | 0 (0) | - |
| Suspected pulmonary infection | 1 (1.6) | 0 (0) | 1.000 |
| Pleural effusion | 5 (7.8) | 5 (8.2) | 1.000 |
| Atelectasis | 11 (17.2) | 12 (19.7) | 0.720 |
| Pulmonary infiltration | 1 (1.6) | 1 (1.6) | 1.000 |
| *Overall PPCs* | 18 (28.1) | 19 (31.1) | 0.711 |

Categorical variables are expressed as the number of patients (%). PACU, post-anesthesia care unit. PPC, postoperative pulmonary complication. Overall PPCs represents the number of patients who had at least one respiratory symptom or sign.

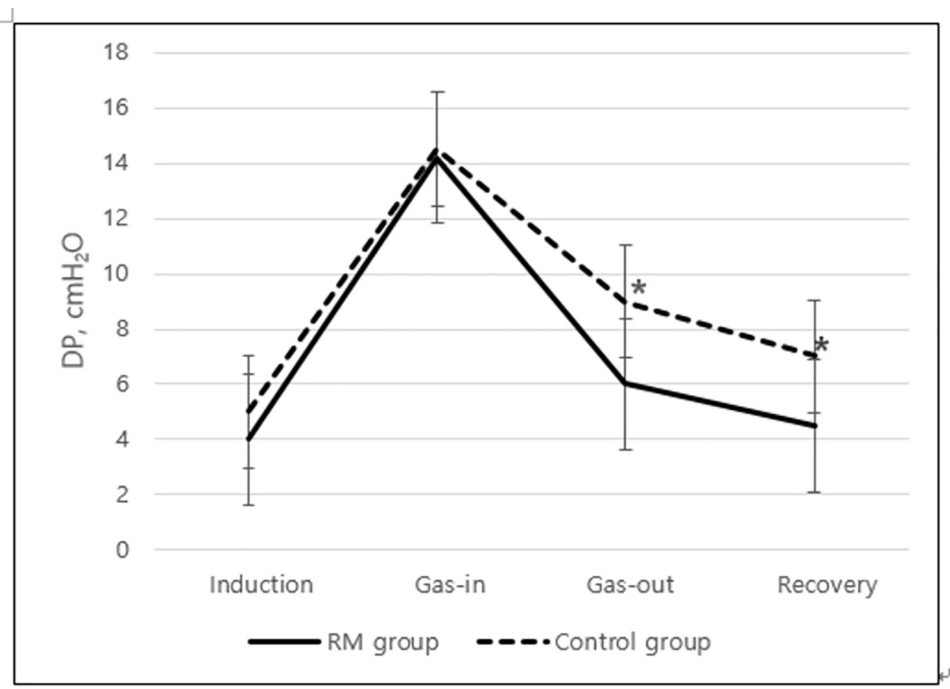

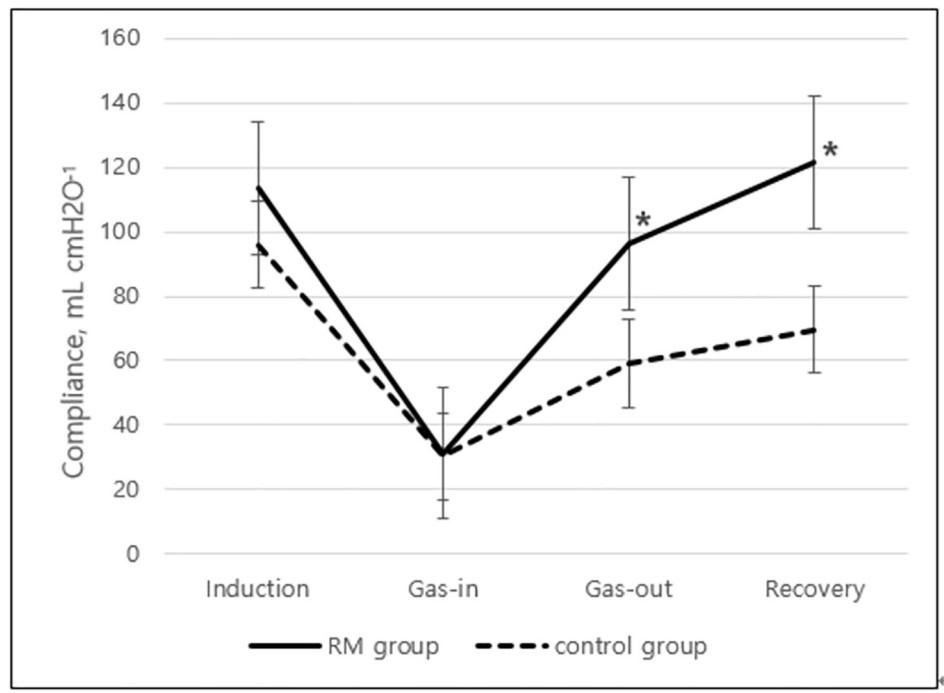

* $P < 0.001$.

**Fig 2. Intraoperative changes in driving pressure and compliance over time.**

prospective observational study conducted in 2017, PIP was the only ventilatory factor associated with a reduction in PPCs in high-risk patients according to the Assess Respiratory Risk In Surgical Patients in Catalonia (ARISCAT) risk score [13].

The use of low TV 4–8 mL kg$^{-1}$, a moderate to high level of PEEP, and the RM during mechanical ventilation is recommended in ARDS patients. However, the RM recommendation is conditional with low to moderate confidence, unlike the strong confidence for low TV and PEEP [7]. The evidence for RM is even lower for perioperative patients undergoing general anesthesia, and various methods have been reported. The RM has been periodically applied every 30 min, one time, or multiple applications at specific time points (i.e., after induction, at the start or end of pneumoperitoneum, at the end of the surgery, and after disconnection from the ventilator), and during postoperative ICU care. A CPAP of 30–40 cmH$_2$O for 30 s, a staircase increase in PEEP, or driving pressure up to the plateau pressure of 30–50 cmH$_2$O or peak pressure of 30–50 cmH$_2$O has been reported [6, 15–17]. Some studies, particularly those in cardiac surgery patients, have reported that a recruited pressure > 40 cmH$_2$O is more effective than a recruited pressure < 40 cmH$_2$O [17]. However, contradictory results of no improvement in PPCs with a recruited pressure > 40 cmH$_2$O during open abdominal surgery, noncardiac general surgery, or laparoscopic surgery have been reported [9, 11, 18]. In contrast, the RM repeated every 30 min with a CPAP of 30 cmH$_2$O for 30 s improves clinical outcomes in patients undergoing major abdominal surgery [10]. Hemodynamic instability and barotrauma are reported complications of the RM and a recent expert panel recommended that the RM be done with the lowest effective pressure and the shortest effective time or the fewest number of breaths [7, 19].

In this study, the RM was performed with a CPAP of 30 cmH$_2$O for 30 s at three critical time points, to encompass the intraoperative and postoperative periods with the minimum number of maneuvers. Intraoperative PIP to achieve the target TV was lower and the calculated compliances were higher in the recruitment group but the incidence of PPCs was comparable between the groups. This result follows previous studies reporting improved intraoperative lung mechanics in the immediate postoperative period but not persisting after tracheal extubation [14, 20, 21]. The RM method we used did not induce hemodynamic instability or any other critical side effect.

This study had several limitations. First, only patients without underlying lung disease were included. However, patients with poor baseline lung function may benefit more from a lung-protective ventilation strategy, considering that they are at higher risk of PPCs. Second, the RM protocol (a pressure of 30 cmH$_2$O and the frequency of RM used) was insufficient for an open lung strategy. Third, a relatively low fixed PEEP of 5 cmH$_2$O may have been inappropriate to maintain the alveolar opening induced by the RM. According to recent studies, an individualized PEEP level minimizes driving pressure, improves intraoperative lung compliance, and reduces postoperative atelectasis [13]. The mean individualized PEEP level under electrical impedance tomography guidance is about 13 cmH$_2$O for laparoscopic surgery and 10 cmH$_2$O for open abdominal surgery, which are considerably higher than our protocol [14]. Fourth, the timing and modality of the lung evaluation could have been less optimized to detect PPCs. More sensitive methods, such as computed tomography and arterial blood gas analysis, could have been used to evaluate the functional status of the lungs and uncover minor evidence of PPCs. Additionally, recent studies that used lung ultrasound reported that the RM significantly reduced intraoperative atelectasis, although the effect was short-lived (i.e., 15 min) after extubation [15]. Thus, the effect of the RM may have not lasted until day 2 after surgery, and earlier imaging would have likely revealed differences.

## Conclusion

Adding the RM with a CPAP of 30 cmH$_2$O for 30 s at the beginning and end of the pneumoperitoneum and immediately before tracheal extubation did not induce any further reduction

in PPCs when low TV and moderate PEEP were applied in patients undergoing laparoscopic colorectal surgery. However, the RM significantly reduced intraoperative PIP and increased the calculated compliance, without posing any significant adverse events during the intervention. Further study is needed to validate the beneficial effect of the RM in patients at increased risk of PPCs.

## Supporting information

**S1 Data.**
(XLSX)

## Author Contributions

**Conceptualization:** Yu Kyung Bae, Sun Woo Nam, Ah-Young Oh, Bo Young Kim, Bon-Wook Koo, Jiwon Han, Subin Yim.

**Data curation:** Yu Kyung Bae, Sun Woo Nam, Ah-Young Oh, Bo Young Kim, Jiwon Han, Subin Yim.

**Formal analysis:** Yu Kyung Bae, Sun Woo Nam, Ah-Young Oh, Bo Young Kim, Jiwon Han, Subin Yim.

**Funding acquisition:** Yu Kyung Bae, Sun Woo Nam, Ah-Young Oh, Subin Yim.

**Investigation:** Yu Kyung Bae, Sun Woo Nam, Ah-Young Oh, Bon-Wook Koo.

**Methodology:** Yu Kyung Bae, Sun Woo Nam, Ah-Young Oh, Bon-Wook Koo, Jiwon Han.

**Project administration:** Yu Kyung Bae, Sun Woo Nam, Bon-Wook Koo, Jiwon Han.

**Resources:** Yu Kyung Bae, Sun Woo Nam, Bon-Wook Koo.

**Software:** Yu Kyung Bae, Sun Woo Nam, Bo Young Kim.

**Supervision:** Yu Kyung Bae, Sun Woo Nam, Ah-Young Oh, Bo Young Kim, Bon-Wook Koo.

**Validation:** Yu Kyung Bae, Sun Woo Nam, Ah-Young Oh, Bo Young Kim, Subin Yim.

**Visualization:** Yu Kyung Bae, Sun Woo Nam, Ah-Young Oh, Bo Young Kim, Subin Yim.

**Writing – original draft:** Yu Kyung Bae, Sun Woo Nam, Ah-Young Oh, Jiwon Han.

**Writing – review & editing:** Yu Kyung Bae, Sun Woo Nam, Ah-Young Oh, Jiwon Han.

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
