## [Decision Letter · Decision Letter 0]

4 Oct 2023

PONE-D-23-24336The effect of alveolar recruitment maneuver during laparoscopic colorectal surgery on postoperative pulmonary complications: A randomized controlled trialPLOS ONE

Dear Dr. Oh,

Thank you for submitting your manuscript to PLOS ONE. After careful consideration, we feel that it has merit but does not fully meet PLOS ONE’s publication criteria as it currently stands. Therefore, we invite you to submit a revised version of the manuscript that addresses the points raised during the review process.

We look forward to receiving your revised manuscript.

Kind regards,

Yeong Shiong Chiew

Academic Editor

PLOS ONE

Journal Requirements:

3. We note that the original protocol that you have uploaded as a Supporting Information file contains an institutional logo. As this logo is likely copyrighted, we ask that you please remove it from this file and upload an updated version upon resubmission.

Additional Editor Comments:

The manuscript has been assessed by our reviewers. There are some feedbacks that we believe would improve the manuscript. Please include a cover letter with a point-by-point response to the comments, describing any changes to the manuscript or rebuttal of any criticisms or requested revisions that you disagreed with.

Reviewers' comments:

Reviewer's Responses to Questions

**Comments to the Author**

1. Is the manuscript technically sound, and do the data support the conclusions?

Reviewer #1: No

Reviewer #2: Partly

Reviewer #3: Yes

Reviewer #4: Yes

Reviewer #5: Yes

2. Has the statistical analysis been performed appropriately and rigorously? 

Reviewer #1: No

Reviewer #2: Yes

Reviewer #3: Yes

Reviewer #4: Yes

Reviewer #5: Yes

3. Have the authors made all data underlying the findings in their manuscript fully available?

Reviewer #1: Yes

Reviewer #2: No

Reviewer #3: No

Reviewer #4: Yes

Reviewer #5: No

4. Is the manuscript presented in an intelligible fashion and written in standard English?

Reviewer #1: No

Reviewer #2: Yes

Reviewer #3: Yes

Reviewer #4: Yes

Reviewer #5: Yes

5. Review Comments to the Author

Reviewer #1: Thanks for letting me review the manuscript entitled " the effect of alveolar recruitment maneuver during laparoscopic colorectal surgery on postoperative pulmonary complications: A randomized controlled trial".

In this study, the authors intend to explore the role of alveolar recruitment maneuver during intraoperative mechanical ventilation, which is a meaningful topic. However, this study has some major defects, such as unreasonable design, unclear key contents, unclear language and logic, so it is not recommended to accept it.

The main shortcomings are as follows:

(1) In low-risk patients undergoing laparoscopic colorectal cancer resection, the incidence of postoperative pulmonary complications is about 30% in this study, which is not in accordance with the clinical practice and is obviously contradictory to the previous studies. This is a design defect and cannot be improved by later modification.

(2) The treatment is not clear described, and there are no references.

(3) The definition of the primary outcome is unclear, and the definition of postoperative pulmonary complications is not clear and unreasonable.

(4) The statistics are obviously insufficient, missing too much cases, not using intention-to-treat analysis, etc.

(5) The authors did not quote the literatures correctly.

Reviewer #2: Overall this was a simple and well conducted trial. There are 3 areas of concern that need to be addressed

1. Clearly, the adequacy of the recruitment manoeuvre (RM) chosen is important to the success of the trial. The authors have chosen a low-level static RM with a Paw of 30 cm H2O for 30 sec but with no justification of this level and with no literature citing that this should be an effective RM in this clinical setting.

In non-operative patients with ARDS, RMs range from stepwise dynamic RMs with PEEP up to 40 cmH2O with tidal PCV up to 55cmH2O for 2 min, to static RMs with Paw 40 cmH2O for 40 sec. The RM in this study is well below this range.

The authors have cited 2 papers that included peri-operative RMs but with no description of the level of RM used to support the use of the RM chosen for use in this study.

Thus it is possible that the RM used was simply not high enough and/or not supported by sufficient PEEP.

The Conclusion in the Abstract and at the end of the paper states “The alveolar recruitment maneuver did not further reduce postoperative pulmonary complications”

This implies that any RMs may not work on this setting.

The Conclusions should specify the level of RM used so it is clear that the results apply to this RM.

“The alveolar recruitment maneuver with Paw 30 cmH20 for 30 sec and PEEP 5 did not further reduce postoperative pulmonary complications”

This should also receive discussion in the section of the paper devoted to study limitations

2. The reduction in PIP in the Recruitment Group has been attributed to improved lung compliance following the RM. This may have been a beneficial effect of the RM, but it may also have been the use of lower tidal volumes in this unblinded study. Clearly the tidal volumes were measured so that they could be set in the range of 6-8ml/kg for recording of the PIPs presented in the paper.

It is important to the results also record the tidal volume and calculated lung compliance at each measurement time before any conclusions in relation to the effects on PIP can be made.

3. The table in relation to PPCs and the overall frequency of PPCs is ambiguous and need clarification. The paper reports a PPC rate of 30%.

It is possible that each complication occurred in a separate patient resulting in 30% of patients having a PCC, OR it is possible that all PCCs occurred in the 17-20% of patients with atelectasis resulting in 18% of patients having a PCC.

The abstract, table and discussion should all also quote the total % of patients who developed a PCC

Reviewer #3: Thank you authors for succesfully completed this research. This research is important to evaluate the efficacy of important intervention which is a pragmatic practice usually performed intraoperatively in the attempt to evaluate this strategy to further reduce post-operative pulmonary complication (PPC) in patients at high risk for PPC.

Overall, authors wrote a comprehensive, detailed explanation regarding the importance of this study, clear research question and methodology which is reproducible by other Anaesthesiologist and clear intention and primary outcome investigated for.

However there are two major issues with regard to the primary outcome:

1. Primary outcome is not clear

 Primary outcome of PPC is not clear as a distinct composite outcome as author presented PPC in Table 3 as separation of PPC into symptoms, signs and clinical feature of PPC rather than PPC stand-alone as a composite outcome as usually being reported by other research in the field of PPC.

-- For example: one patient could have Hypoxemia in PACU, as well as Pleural Effusion in the ward later on, thus the question will arise as whether this patient considered as 1 PPC or 2 PPC? as per convention and usual practice, PPC in this case is considered only 1 incidence despite having 2 clincal signs at two different time in the same patient.

-- Further explanation regarding PPC as a composite outcome rather than separate distinct clinical signs can be considered here in https://doi.org/10.1093/bja/aex002 and example of composite outcome of PPC reported can be found here in Table 3 in this study here https://doi.org/10.1371/journal.pone.0274749

2. Please include line chart indicating the changes of pressure over time with regarding to Peak inspiratory pressure in both group and compare from initial time at induction, gas-in, gas-out and recovery changes and compare with proper statistical anlysis, such as Repeated Measures ANOVA (RMANOVA) to study the significance changes drop over time for within the group and comparative statistical changes between the group over the time course using RMANOVA. this test is more appropriate than student t-test used for table 2

There are some other minor issues such as:

1. Table 1 presented baseline data without statistical analysis comparison between the two group to provide the readers to study and ensure both groups have comparable baseline characteristic, thus permitting comparison of intended primary outcome as the baseline was similar in both group

2. Safety assessment of intervention of alveolar recruitment in this study need to be presented as this was planned based on the research proposal provided: "If low blood pressure occurs during the study, general treatment such as intravenous fluid administration and administration of vasopressors will be administered at the discretion of the anesthesiologist in charge. If serious complications occur due to ventilation therapy after surgery, immediately report them to the Seoul National University Bundang IRB in writing via phone or email"

 please present any incidence of BP low or require IV fluid or vasopressor during or due to alveolar recruitment intervention

3. Allocation ratio into groups intervention and control was not mentioned, but it was implied to be 1:1. please mention in the text

4. In addition to or instead of Peak Inspiratory Pressure, please consider including driving pressure calculated by (Peak Insp minus PEEP) as part of secondary outcome as many research and convention and consensus now studying effect of driving pressure to multiple various outcome in mechanical ventilation

5. Primary outcome table should come before secondary outcome table. Table 2 and 3 should switch the order

Reviewer #4: Nice attempt to explore the intervention. Methodologically sound but not novel to this field and provides no new insight or information on this topic. I commend the effort but this study does not add any value to the wealth of of knowledge on this topic. The design of the study should be re-looked and the PEEP strategy needs refinement to meet better end-points i.e. PEEP titration to compliance, blood gases or EIT

I feel this study is not clinically adding any insights to the field and should be re-examined for relevance. It would be more relevant if some other tool such as electrical impedance tomography(EIT) determined what the optimal PEEP should be for the individual patients.

Reviewer #5: Manuscript PONE-D-23-24336 is a randomized controlled trial on perioperative alveolar recruitment and its effect on postoperative pulmonary complications. The study is well performed and written. Analysis is sound and data support the conclusions.

Comments.

1. In table 2 PIP is expressed in cmH2O, in the legend mmHg is indicated.

2. Page 14, last sentence of first paragraph: 'Nevertheless, a reduction ....... risk of PCCs.' This remark is not supported by the data.

3. Page 15, why was a high FiO2 of 0,8 -1,0 used in anesthesia recovery? Is this protocollized? Why not titrated on saturation? I presume that, with a base fiO2 of 0,4 during anesthesia, FiO2 is perioperatively titrated. The authors should reflect on this issue.

4. The authors should report the side effects of the recruitment maneuver in the studied patients.

6. PLOS authors have the option to publish the peer review history of their article (what does this mean?). If published, this will include your full peer review and any attached files.

Reviewer #1: No

Reviewer #2: **Yes: **Prof David V Tuxen

Reviewer #3: **Yes: **Abdul Jabbar Ismail

Reviewer #4: **Yes: **MGA Miller

Reviewer #5: No

---

## [Author Response · Author response to Decision Letter 0]

11 Jan 2024

We have revised the manuscript based on the reviewers' suggestions. 

In order to present the research results more clearly, we replaced tables with figures during the revision. Additionally, new statistical analyses were included in the revision. In the conclusion section, we made further modifications to better elucidate the purpose of this study. We have also made adjustments to references, including additions and removals. 

The manuscript has undergone English language corrections as well. 

We express our gratitude once again to the reviewers who dedicated their time to reviewing this research. We look forward to positive outcomes.

---

## [Decision Letter · Decision Letter 1]

19 Feb 2024

PONE-D-23-24336R1복강경 대장 수술을 받는 환자에서 폐포모집요법이 수술 후 폐합병증 발생에 미치는 영향PLOS ONE

Dear Dr. Oh,

Thank you for submitting your manuscript to PLOS ONE. After careful consideration, we feel that it has merit but does not fully meet PLOS ONE’s publication criteria as it currently stands. Therefore, we invite you to submit a revised version of the manuscript that addresses the points raised during the review process.

We look forward to receiving your revised manuscript.

Kind regards,

Yeong Shiong Chiew

Academic Editor

PLOS ONE

Journal Requirements:

Additional Editor Comments (if provided):

The manuscript has been assessed by our reviewers. There are a few feedbacks that we believe would improve the manuscript. Please address them by including a cover letter with a point-by-point response to the comments. In addition, the authors are reminded to include English version of the Title and Abstract in the editorial manager system. They should be similar to the revised manuscript.

Reviewers' comments:

Reviewer's Responses to Questions

**Comments to the Author**

1. If the authors have adequately addressed your comments raised in a previous round of review and you feel that this manuscript is now acceptable for publication, you may indicate that here to bypass the “Comments to the Author” section, enter your conflict of interest statement in the “Confidential to Editor” section, and submit your "Accept" recommendation.

Reviewer #3: All comments have been addressed

Reviewer #4: All comments have been addressed

Reviewer #5: All comments have been addressed

2. Is the manuscript technically sound, and do the data support the conclusions?

Reviewer #3: Yes

Reviewer #4: Yes

Reviewer #5: (No Response)

3. Has the statistical analysis been performed appropriately and rigorously? 

Reviewer #3: Yes

Reviewer #4: Yes

Reviewer #5: (No Response)

4. Have the authors made all data underlying the findings in their manuscript fully available?

Reviewer #3: Yes

Reviewer #4: Yes

Reviewer #5: (No Response)

5. Is the manuscript presented in an intelligible fashion and written in standard English?

Reviewer #3: Yes

Reviewer #4: Yes

Reviewer #5: (No Response)

6. Review Comments to the Author

Reviewer #3: Thank you for responding well to suggestion provided during previous review.

Only several minor revision:

1. in Line 40, i think there is no "dot" after the standard deviation and should be a continuous sentence. please confirm

2. Please add in abstract in line 45 that the RM did not had any significant side effect (addresing "safety" of maneuvre using short few words") : "undergoing laparoscopic colorectal surgery and was not associated with any significant adverse events"

3. in line 218: please simplify this sentence: suggestion: "The importance of preventing PPCs must be emphasized, given that.." (the use of cannot be overemphasized is confusing for certain english reader, as this sentence followed by emphasizing the effect of prolonged LOS and expenses)

3. in line 222, please add driving pressure findings as well as it has been presented in result section. can be just simply: "...significantly reduce PIP and DP...."

4. please add in the safety aspect of RM performed with short few words in line 281 after the "calculated compliance": suggestion: "......increased the calculated compliance, without posing any significant adverse events during the intervention"

Reviewer #4: Thank you for addressing the comments. I am satisfied with the revisions made.

The authors have addressed all the relevant issues that were highlighted.

Reviewer #5: (No Response)

7. PLOS authors have the option to publish the peer review history of their article (what does this mean?). If published, this will include your full peer review and any attached files.

Reviewer #3: **Yes: **Abdul Jabbar Ismail

Reviewer #4: **Yes: **Malcolm G.A. Miller(MD)

Reviewer #5: **Yes: **Dr. Dennis CJJ Bergmans, MD, PhD

---

## [Author Response · Author response to Decision Letter 1]

23 Feb 2024

After reviewing the references, we found them to be appropriate.

This study did not receive any funding from our institution. 

Based on the feedback from the reviewers, we have made revisions to the manuscript.

We've included some of the suggestions that reviewer offered in our manuscript.

---

## [Decision Letter · Decision Letter 2]

4 Apr 2024

PONE-D-23-24336R2Effect of the alveolar recruitment maneuver during laparoscopic colorectal surgery on postoperative pulmonary complications: A randomized controlled trialPLOS ONE

Dear Dr. Oh,

Thank you for submitting your manuscript to PLOS ONE. After careful consideration, we feel that it has merit but does not fully meet PLOS ONE’s publication criteria as it currently stands. Therefore, we invite you to submit a revised version of the manuscript that addresses the points raised during the review process.

We look forward to receiving your revised manuscript.

Kind regards,

Yeong Shiong Chiew

Academic Editor

PLOS ONE

Journal Requirements:

Reviewers' comments:

Reviewer's Responses to Questions

**Comments to the Author**

1. If the authors have adequately addressed your comments raised in a previous round of review and you feel that this manuscript is now acceptable for publication, you may indicate that here to bypass the “Comments to the Author” section, enter your conflict of interest statement in the “Confidential to Editor” section, and submit your "Accept" recommendation.

Reviewer #6: (No Response)

2. Is the manuscript technically sound, and do the data support the conclusions?

Reviewer #6: Yes

3. Has the statistical analysis been performed appropriately and rigorously? 

Reviewer #6: Yes

4. Have the authors made all data underlying the findings in their manuscript fully available?

Reviewer #6: Yes

5. Is the manuscript presented in an intelligible fashion and written in standard English?

Reviewer #6: Yes

6. Review Comments to the Author

Reviewer #6: A two-arm randomized control study was conducted which aimed to compare the clinical and biological postoperative pulmonary complications up to 7 days post surgery. Surgical ventilation was adjusted to maintain a tidal volume of 6–8 ml kg -1 and a positive end-expiratory pressure of 5 cmH2O in the experimental arm. The overall incidence of postoperative complications was not significantly different between the groups.

Minor revisions:

1- Line 145: Indicate the statistical testing method which achieves 80% power. Perhaps it is the chi-square test for comparing proportions.

2- Line 151: Indicate the underlying covariance structure used in the linear mixed model and the criteria for selecting it.

3- Table 1: Indicate the percentages of males (or females) in each group in addition to the frequencies.

4- Line 188: State the compliance rates.

5- Line 203: Express p-values more precisely, rather than p>0.05.

7. PLOS authors have the option to publish the peer review history of their article (what does this mean?). If published, this will include your full peer review and any attached files.

Reviewer #6: No

---

## [Author Response · Author response to Decision Letter 2]

11 Apr 2024

We have further strengthened the statistical explanation of our manuscript.

Based on the feedback from the reviewers, we have made revisions to the manuscript.

We greatly respected the reviewers' opinions and reflected them in our manuscript.

---

## [Editor Report · Decision Letter 3]

16 Apr 2024

Effect of the alveolar recruitment maneuver during laparoscopic colorectal surgery on postoperative pulmonary complications: A randomized controlled trial

PONE-D-23-24336R3

Dear Dr. Oh,

We’re pleased to inform you that your manuscript has been judged scientifically suitable for publication and will be formally accepted for publication once it meets all outstanding technical requirements.

Kind regards,

Yeong Shiong Chiew

Academic Editor

PLOS ONE
---

## [Editor Report · Acceptance letter]

26 Apr 2024

PONE-D-23-24336R3 

PLOS ONE

Dear Dr. Oh, 

I'm pleased to inform you that your manuscript has been deemed suitable for publication in PLOS ONE. Congratulations! Your manuscript is now being handed over to our production team.

Kind regards, 

on behalf of

Dr. Yeong Shiong Chiew 

Academic Editor

PLOS ONE